# Sievert or Gray: Dose Quantities and Protection Levels in Emergency Exposure

**DOI:** 10.3390/s23041918

**Published:** 2023-02-08

**Authors:** Chiara Ferrari, Guglielmo Manenti, Andrea Malizia

**Affiliations:** 1Department Industrial Engineering, University of Rome Tor Vergata, Via del Politecnico 1, 00133 Rome, Italy; 2Department of Biomedicine and Prevention, University of Rome Tor Vergata, Via Montpellier 1, 00133 Rome, Italy

**Keywords:** radiation emergency, dose quantities, dose reference levels

## Abstract

Mitigation or even elimination of adverse effects caused by ionizing radiation is the main scope of the radiation protection discipline. The interaction of radiation with living matter is quantified and correlated with biological effects by dose. The Sievert is the most well-known quantity, and it is used with the equivalent and effective dose to minimize stochastic effects. However, Gray is the reference quantity for sizing tissue reactions that could occur under high-exposure conditions such as in a radiation emergency. The topics addressed in this review are the choice to move from Sievert to Gray, how the operational quantities for environmental and individual monitoring of the detectors should consider such a change of units, and why reference levels substitute dose levels in emergency exposure.

## 1. Introduction

The scope of the radiation protection discipline is to mitigate or even eliminate the harmful effects caused by ionizing radiation on the human population and to preserve the environment. The interaction of radiation with living matter must be sized and correlated with biological effects to address the problem.

Dosimetry measures the energy transmitted to the matter, the absorbed dose, based on changes induced by ionizing radiation. However, this is insufficient to quantify the health detriment to living beings. The biological effects depend on the total amount of absorbed energy and many other properties of the radiation field and targeted tissues. A high level of absorbed dose induces tissue damage due to cell death (tissue effects). In addition, every level can provoke cumulative effects (stochastic effects).

### 1.1. Tissue Reactions

The absorption of radiation above a certain threshold damages living tissue and causes deterministic effects. The severity of these effects is proportional to the energy absorbed over the mass, measured in Gray: energy deposited per unit mass J kg^−1^, as stated by the International System of Units (SI). It is a physically measurable quantity.

To calculate organ damage, the absorbed dose is multiplied by the coefficient Relative Biological Effectiveness (RBE), which considers the type of radiation field, the biological effect reported, the duration of exposure, and the tissue involved. The International Commission on Radiation Protection (ICRP) states that the use of the absorbed dose weighed with RBE of an organ/tissue D_RBE_ should be publicly declared, using the same units of the absorbed dose Gray (Gy) [1]. On the contrary, the US National Radiological Protection Board (NRPB) suggests using the Gray Equivalent Unit (Gy-Eq) [2]. In either case, this formalism allows the addition of doses from different types of radiation; this is a valuable advantage for quantifying the overall toxic effects caused by mixed particle radiation fields and different sources of exposure (internal and external).

### 1.2. Stochastic Effects

All radiation doses can cause stochastic effects, whose likelihood is dose-related; the severity, on the other hand, is dose-independent. According to the Linear No-Threshold Model (LNT), there is no lower dose limit for the occurrence of toxic effects. When dealing with stochastic effects, the reference unit is Sievert [Sv]. All regulatory limiting values and most of the reference values are expressed in Sievert. Sievert, introduced in 1954 upon the recommendation of ICRP recommendation (former Radiation Equivalent Man, 1 rem = 0.01 Sv), is a radiobiological quantity defined for humans, used for “equivalent organ dose” and “effective dose” quantities [3].

The equivalent dose is the dose to the organ/tissue multiplied by a factor determined by the properties of the radiation impinging, w_R_. The w_R_ factors depend on the linear transfer energy (LET) of radiation, discontinuing the use of RBE in 1991.

The effective dose quantifies the risk due to the dose absorbed throughout the body. The result is the sum of the equivalent doses for organ/tissue, weighted by the tissue weighting factor w_T_. The w_T_ values were modified by ICRP 26 [4], ICRP 60 [3], and ICRP 103 [5], the latest of which suggests the values used considering the new epidemiological evidence and detriment principles. They are evaluated on a standard human body, therefore disregarding individual characteristics such as gender, age, the actual size of organs, and body, as well as individual physiological characteristics. The formalism of the effective dose has the great advantage of allowing the summation of internal and external exposure, allowing the comparison of different types of exposure through the same metric.

The effective dose concept was historically developed to compare the risks of occupational radiation exposure to other occupational risks. Later, it was also adopted to assess radiation protection limits for the general population and then to compare the risks of medical exposures. Despite the great success and diffusion among radiation professionals, the effective dose concept is often not always correctly understood because it does not quantify the risk to an individual [6]. Moreover, the use of Sievert for the equivalent and effective doses generates confusion. In fact, ICRP suggests discontinuing the use of equivalent dose in dose limits to avoid eye lens and skin tissue reactions because they are meant to protect against deterministic effects.

### 1.3. Operational Quantities

The protection quantities are not measurable, but operational field quantities or numerical models allow their estimation, according to the ICRU formalism, Figure 1.

Currently, in a routine situation, under a well-regulated radiation protection system, the Sievert is used for dose assessment for all categories of exposure (workers, patients, and members of the public); the Gray is used in high-dose medical exposures. For example, in radiotherapy treatments, risk/benefit and therapeutic dose are measured with Gray or RBE weighted Gray [8]; The prevention of skin tissue reaction in interventional radiology and CT examination uses Peak Skin Dose (PSD) and Computed Tomography Dose Index (CTDI), measured in Gray [9].

In a radiation emergency, in nonroutine situations, or in events that necessitate immediate action to mitigate a radiological hazard or its adverse effects, both units must be used.

To meet all these requirements, various sensors are designed according to the scope, and different calibration curves are available depending on the dose unit requested.

## 2. From Sievert to Gray

When can we consider an acceptable change of units from effective dose to absorbed dose or RBE-weighted absorbed dose of an organ/tissue? What is the quantity of radiation that justifies considering stochastic effects or also considering tissue effects?

The effective and equivalent dose should be avoided to drive the clinical judgment for tissue damage, i.e., overexposure. The right quantity is the absorbed dose or the absorbed dose weighted by RBE. Great attention must be paid to RBE values that are usually lower than the w_R_. The inappropriate use of equivalent organ dose to estimate the deterministic effects could lead to overestimating the radiobiological effect.

The threshold value of the deterministic effects is around 100 mGy, low-LET, or hight-LET radiation independently. The same threshold accounts for statistically significant cancer radiation induction, despite increasing evidence for doses less than 100 mGy [10]. As the absorbed dose increases above 100 mGy, the change from Sievert to Gray becomes necessary due to the increasing relevance of tissue damage. As a baseline, the rule of thumb for planned medical exposure can be useful; when the effective dose approaches 100 mSv, some tissues probably have already reached 100 mGy. However, one should be aware that in inhomogeneous exposures the effective dose may remain low.

The effective dose is appropriate up to 1000 mSv (1 Sv) only in emergency exposure situations. The aim is not to introduce further difficulties in changing measurement units, as should be expected when dealing with acute exposure because radiation protection monitoring instruments are usually calibrated in terms of the operational quantity Ambient Dose Equivalent, H*(p), and the personal monitoring dosimeters in terms of Personal Equivalent Dose, Hp(d). However, two fundamental concepts must not be forgotten: tissue reaction potential, especially from strong nonuniform radiation fields, and the increase in the nominal cancer risk coefficient when the tissue dose is greater than 100 mGy by low LET radiation and the dose rate is greater than 5 mGy/h [1]. During an emergency, the effective dose plays a role, but a comprehensive risk assessment requires an estimation of the organ/tissue dose of the individuals exposed.

Focusing on the operational quantities used by the monitoring instrumentation, the ICRU recently presented new definitions in the case of external radiation, improving the methodology for estimating the effective dose and promoting its gradual adoption [7]. The influence of these novelties must be investigated from the point of view of radiation emergency monitoring: the new operational quantities should allow for a more accurate estimation of the effective dose and dose to the extremities, but relevant differences remain for some photon and neutron fields [11,12,13].

## 3. Protection Criteria in an Emergency: From Dose Limits to Reference Values

During an emergency, the first responders, the advisors of decision makers (considered emergency workers), and the population (the civilians) deal with different risk criteria. Additionally, the regulatory dose limits are no longer valid. Using the “reference level” instead of the “dose limit” introduces flexibility when facing an emergency since it is more suitable for real needs [14]. The concept of a reference level is also used for existing exposure and medical exposures, where limit values are ineffective, but work for optimization should not stop.

Tissue reaction prevention is still a cornerstone of radiation protection, but it could be considered not of primary importance during emergencies: the dose absorbed by a rescuer should have no definite limit if the overall benefit justifies the potential harm to each person’s health; it is the case of life-saving interventions by informed volunteers. According to the ICRP, effective doses below 1000 mSv do not cause severe deterministic effects; below 500 mSv should avoid other deterministic effects. Therefore, exposure during urgent rescue operations should not exceed 1 Sv.

Unlike the countermeasures for the population, they apply to the prevention of stochastic effects at low doses or low dose rates, setting the levels of emergency planning typically in the range of 20–100 mSv per year.

Table 1 reports on the ICRP 103 scheme of protection criteria for emergency exposure. Occupational and public exposures refer to the rescuer and the general population, respectively. The table is a declaration of radiation protection rules as guiding principles. Their operational implementation in accordance with observable quantities is the task of bodies such as the International Atomic Energy Agency (IAEA).

Intervention levels, according to ICRP 60 [3], are characterized by the doses that are avoided due to specific countermeasures; they are useful when the decision makers are developing a protection strategy to optimize individual countermeasures, as well as supplementing reference levels used to evaluate protection strategies. The ICRP means that intervention levels are the doses expected to be incurred after protective measures have been discontinued. The generic criteria in the series of IAEA EPR reports are a further development of these.

Accordingly, the IAEA reviewed Basic Safety Standards [15]. As a result, the European Union issued the Council Directive 2013/59/Euratom [16] based on the recommendations of the ICRP and in close cooperation with the IAEA, providing guidance for the creation of an emergency response plan, based on which of the EU Member States were required to develop national implementing laws.

As an example of non-EU national implementation, the US National Council of Radiation Protection (NCRP) issued the report n°180 [2], assuming the same goals of the ICRP, with some differences with respect to the EU directive. It accounted for five categories of exposures adding nonhuman biota and emergency workers. The emergency worker is classified as a transitory exposure category that begins after a radiological or nuclear emergency [2,16].

## 4. Guidance Values in Occupational Exposure—The Emergency Worker

The IAEA reports of the Emergency Preparedness and Response (EPR) series are based on the ICRP reference. The guidance values (reference levels) for occupational exposures in an emergency recall the operational quantity Hp (10), the estimator of strongly penetrating external radiation due to an isotropic radiation field. The implied concept is that every effort was made to protect against external exposure due to skin contamination.

Table 2 contains the IAEA guidance values for managing emergency worker doses. The IAEA General Safety Requirements contain a deeper examination considering the necessity of evaluating the organ/tissue dose from all exposure pathways, including the radionuclide intake; therefore, effective dose E and RBE-weighted dose AD_T_ reference values are included. Different radiation doses characterize the guidance according to diverse tasks, ranging from action to averting collective dose (the lowest) to life-saving action (the highest). Once again, the relevance of risk awareness of rescuers is underlined in the most challenging task.

According to US regulations, the emergency worker is defined as one of the “categories of exposure”, different from ICRP, which considers them a case of occupational exposure. The NRPB report provides “numerical protection criteria” (the reference levels) for emergency workers that should not be exceeded. Table 3 shows the numerical protection criteria for emergency workers. Two exposure situations (tasks) determine two different numeric protection criteria. Over 100 mSv, the units change from Sievert to Gray; the quantity refers to the cumulative dose absorbed by the whole body, therefore emphasizing the occurrence of deterministic effects. The 0.5 Gy numeric level is a “decision dose” because the command must take the necessary decision as the value approaches. It is clearly specified that the proposed values are not “limits” because initial doses to individuals may exceed the applicable numeric protection criterion in some exposure situations. In fact, these values do not represent a border between safe and unsafe.

## 5. Generic Criteria and Guidance Quantities in Population Exposure

For the general population, the IAEA scheme proposes immediate protective actions according to organ dose reference levels or overall dose levels to ensure the recommendations of the ICRP for preventing both acute and stochastic effects. In addition, it includes criteria for developing the operational levels needed for decision making on protection and response actions.

The amount of radiation dose received or projected is the reference, considering that all exposure can increase the stochastic effect, and acute exposure can result in deterministic tissue effects. The received dose is the actual absorbed dose; the projected dose could be avoided or reduced by taking immediate preventive protective measures.

The RBE-weighted dose AD*_T_* and the committed RBE-weighted dose of an organ or tissue AD*_T_* (Δ) are the quantities used to manage acute exposition. Their definition allows the addition of doses from different kinds of radiation and energy, for instance, from mixed photon and neutron fields, as could occur in the event of a fission reactor emergency or for offensive use of nuclear devices.

The generic dose criteria and subsequent actions to be taken in acute exposition are presented in Table 4. Projected or absorbed doses trigger different actions.

The reduction of the stochastic risk requires the assumption of urgent and protective actions that are graduated according to equivalent and effective dose criteria into the Table 5. The targeted organs are the thyroid and the fetus.

The guidance for a ready response is given according to operational criteria, i.e., measurable, or observable quantities: operational intervention levels (OILs) and emergency action levels (EALs) [18]. The OILs are gamma, beta, and alfa surface contamination levels of detailed surfaces or volumes (ground surfaces, skin, and milk/food/water) for survey measurements. The EALs are specific, predetermined, observable operational criteria used to detect, recognize, and classify an event at facilities.

## 6. Conclusions

The Sievert unit is operationally useful if an expert is dealing with stochastic risk mitigation, i.e., in case of low level of exposure, for assessing compliance with radiation protection limits. Despite the great diffusion of this unit of measure, its comprehension is often incomplete.

If we move from limiting stochastic effects to managing acute exposures, the use of correct quantities is a matter of paramount importance. As a result, in addition to Sievert, Gray and RBE weight factors must be considered. Stochastic risk and deterministic effects measures should be used for estimated doses above 100 mSv or absorbed doses to organ/tissue greater than 100 mGy. As the absorbed doses increase, tissue damage becomes the prevalent effect. However, only for emergency exposures, the use of the Sievert unit is allowed up to 1 Sv, to avoid uncertainties along with the real-time event evolution. Therefore, the operational quantities of Ambient Dose Equivalent H*(d) and Personal Dose Equivalent Hp(d) are still considered. The use of new operational quantities proposed by ICRU 95 for external monitoring should be investigated to understand how and if they could affect the calibration coefficients and the detectors.

Dose levels given in an emergency do not have the same meaning as in the case of exposure under the regulations. The proposed reference levels and numerical protection criteria are not limiting values. They help to lower exposure because optimization efforts should never stop. However, they can be overcome in exceptional circumstances.

The IAEA EPR series reports are based on ICRP statements and give operational values in the event of an emergency involving rescuers and the population. For emergency workers, the operational quantities “personal dose equivalent” Hp(10), “effective dose” E, and “RBE-weighted absorbed dose to a tissue or organ” ADT is used to define the reference values. For the population, stochastic risk and tissue effects are managed using Sievert and Gray units to define different intervention levels. The measurable or observable quantities that trigger protective action of intervention levels are operational intervention levels (OILs) and/or emergency action levels (EALs).

## Figures and Tables

**Figure 1 sensors-23-01918-f001:**
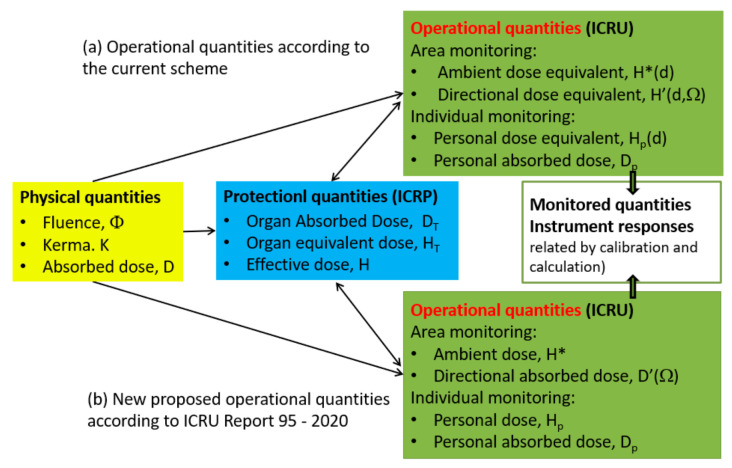
Relationship between the protection quantities of ICRP Publication 103 [5] and the operational ones of ICRU according to (**a**), the current scheme; (**b**) Report 95 [7], the new proposed operational quantities.

**Table 1 sensors-23-01918-t001:** Protection criteria in emergency exposure situations, from ICRP 103 [5]—see Appendix A for a,b,c,d’s explanations.

Category of Exposure	ICRP 103 Recommendations
*Emergency exposure situations*	*Reference levels ^a,b^*
**Occupational Exposure**	
- live-saving (informed volunteers)	No dose restrictions if benefit to others outweighs rescuer’s risk ^c^
- other urgent rescue operations	1000 or 500 mSv ^c^
- other rescue operations	≤100 mSv ^c^
**Public Exposure**	
- all countermeasures combined in an overall protection strategy	In planning, typically between 20 and 100 mSv/year according to the situation ^d^

**Table 2 sensors-23-01918-t002:** Guidance values for restricting exposures of emergency workers from IAEA [17]—see Appendix B for a,b,c,d’s explanations.

Task	Guidance Value ^a^		
	Hp(10) ^b^	E ^c^	AD_T_ ^d^
- Life-saving actions.	<500 mSv	<500 mSv	<1/2 AD_T_
	This value can be exceeded in circumstances in which the expected benefits for others clearly overestimate the health risks of the emergency worker, and the emergency worker volunteers to take the actions and understands and accepts these health risks.		
- Actions to prevent severe deterministic effects and actions to prevent the development of catastrophic conditions that could significantly affect people and the environment.	<500 mSv	<500 mSv	<1/2 AD_T_

**Table 3 sensors-23-01918-t003:** Numeric protection criteria for the management of dose to an emergency worker, NCRP [2]—see Appendix C for a,b’s explanations.

Exposure Situation	Numeric Protection Criteria (mSv) ^a^ (Effective Dose, Except Where Noted)	Suitable for Application as a Regulatory Limit
**Exposure of emergency workers**		
- During lifesaving activities or actions to prevent catastrophic situations. Includes other urgent rescue activities.	Managed by a decision dose ^b^ of 0.5 Gy (cumulative whole-body absorbed dose), implemented at the command level.	No
- For other emergency activities, including extended activities after initial lifesaving, rescue, and damage control response.	Should not exceed 100 mSv for the duration of the emergency operation; reduce using optimization.	No

**Table 4 sensors-23-01918-t004:** Tissue effects absorbed (projected or received) dose levels that require immediate protective actions, from IAEA [17]—see Appendix D for a,b,c,d,e,f,g,h,i’s explanations.

Generic Criteria		Example of Protective Actions and Other Response Actions
**External acute exposure (<10 h)**		If the dose is projected:
- AD_red marrow_ ^a^	1 Gy	- Take precautionary urgent protective actions immediately (even under difficult conditions) to keep doses below the generic criteria. - Provide public information and warnings. - Curry out urgent decontamination.
- AD_fetus_	0.1 ^b^ Gy	
- AD_tissue_ ^c^	25 Gy at 0.5 cm	
- AD_skin_ ^d^	10 Gy to 100 cm^2^	
**Internal acute from acute intake (D = 30 days ^e^)**		If the dose has been received:
- AD(D)_red marrow_	0.2 Gy for radionuclides with Z ≥ 90 ^f^ 2 Gy for radionuclides wit Z ≤ 89 ^f^	- Perform immediate medical examination, consultation, and indicated medical treatment. - Carry out contamination control, - Carry out immediate de-corporation ^g^ (if applicable). - Carry our registration for long term health monitoring (medical follow up). - Provide comprehensive psychological counseling.
- AD(D)_tyroid_	2 Gy	
- AD(D)_lung_ ^h^	30 Gy	
- AD(D)_colon_	20 Gy	
- AD(D)_fetus_ ^i^	0.1 Gy	

**Table 5 sensors-23-01918-t005:** Stochastic risks, dose levels that require immediate protective actions, from IAEA [17].

Generic Criteria		Example of Protective Actions and Other Response Actions
**Projected dose that exceeds the following generic criteria: take *urgent* protective** **actions and other response actions**		
H_thyroid_	50 mSv in the first 7 day	Iodine thyroid blocking
E	100 mSv in the first 7 days	Sheltering; evacuation; decontamination; restrictions of
H_fetus_	100 mSv in the first 7 days	food, milk, and drinking water; contamination control; reassurance of the public H
**Projected dose that exceeds the following generic criteria: take *early* protective actions and other response actions**		
E	100 mSv in the first year	Temporary relocation; decontamination; restrictions
H_fetus_	100 mSv in the first year	on food, milk and drinking water; reassurance of the public
**Dose that has been received and that exceeds the following generic criteria: take *longer term medical actions* to detect and to effectively treat radiation induced health effects**		
E	100 mSv in a month	Health screening based on equivalent doses to specific radiosensitive organs (as a basis for medical follow-up); counseling
H_fetus_	100 mSv for the full period of in utero development	Counseling to allow informed decisions to be made in individual circumstances

Note: H_T_—equivalent dose in an organ or tissue T (in this table there are “thyroid” and “fetus”); E—effective dose.

## Data Availability

Not applicable.

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
