# Peer review of "Sievert or Gray: Dose Quantities and Protection Levels in Emergency Exposure"

_sensors, 2023, doi:10.3390/s23041918_

Round 1
Reviewer 1 Report
Reviewer comments for authors:
The authors analyzed the application of new operational quantities according to ICRU 95 recommendations.
Shorten the Introduction to avoid unnecessary repetition.
Table 1 refers to data from ICRP103, not 107 as stated on page 4.
Most of the tables are robust, so I suggest the authors reorganize the text and move those Tables to an appendix.
Please cite the article of M. Caresana et al. titled as “Impact of new operational dosimetric
quantities on individual monitoring services” published in J. Radiol. Prot.( 41 (2021) 1110–1121).
Reviewer 2 Report
Dear authors,
The submitted manuscript presents a review of the latest scientific reports of international bodies regarding dose values physical quantities, however I have some concerns, expressed below in the form of comments.
Comment 1. There is not any new information in the submitted manuscript. As such I cannot consider it as a research paper. However, it is might be a useful review paper. Please stress the word "review" in the abstract and introduction section.
Comment 2. Abstract the authors state that "But during an emergency, it could not fit for the scope because tissue effects can take place, and they are quantified with the Gray unit.". Tissue effects are considered through wT and this is the sievert unit, since emergency effects are usually whole-body effects. Of course in special cases, as mentioned in the anuscript, the RBE may be considered. The authors are kindly asked to rewrite the sentence.
Comment 3.The authors state that "The aim is not to introduce further difficulties in changing measure units, as should be expected when dealing with acute exposure. The aim is not to introduce further difficulties in changing measure units, as it should" Please re write these sentences, it seems like a repetitiion.
Comment 4. References at the sentences end of Table 4 and 5 legends. Reference 9 is the EU council directive not the IAEA. Please correct appropriately.
Comment 5. The authors should insert information (eg a table or text) with examples that may be considered an emergency
with kind regards
Round 2
Reviewer 2 Report
Dear authors
I am OK with your corrections
thank you